# Integrated treatment of hepatitis C virus infection among people who inject drugs: A multicenter randomized controlled trial (INTRO-HCV)

**Lars T. Fadnes**[1,2]*, **Christer Frode Aas**[1,2], **Jørn Henrik Vold**[1,2], **Rafael Alexander Leiva**[3], **Christian Ohldieck**[1], **Fatemeh Chalabianloo**[1,2], **Svetlana Skurtveit**[4,5], **Ole Jørgen Lygren**[1,6], **Olav Dalgård**[7,8], **Peter Vickerman**[9], **Håvard Midgard**[7,10], **Else-Marie Løberg**[1,11,12], **Kjell Arne Johansson**[1,2], **for the INTRO-HCV Study Group**¶

1 Bergen Addiction Research, Department of Addiction Medicine, Haukeland University Hospital, Norway, 2 Department of Global Public Health and Primary Care, University of Bergen, Norway, 3 Department of Medicine, Haukeland University Hospital, Norway, 4 Norwegian Centre for Addiction Research, University of Oslo, Norway, 5 Department of Mental Disorders, Norwegian Institute of Public Health, Oslo, Norway, 6 ProLAR Nett, Norway, 7 Department of Infectious Diseases, Akershus University Hospital, Lørenskog, Norway, 8 Institute for Clinical Medicine, University of Oslo, Norway, 9 Population Health Sciences, Bristol Medical School, University of Bristol, United Kingdom, 10 Department of Gastroenterology, Oslo University Hospital, Norway, 11 Department of Clinical Psychology, University of Bergen, Norway, 12 Division of Psychiatry, Haukeland University Hospital, Norway

¶ Membership of INTRO-HCV Study Group is provided in the Acknowledgments.
* lars.fadnes@uib.no

## Abstract

### Background

The standard pathways of testing and treatment for hepatitis C virus (HCV) infection in tertiary healthcare are not easily accessed by people who inject drugs (PWID). The aim of this study was to evaluate the efficacy of integrated treatment of chronic HCV infection among PWID.

### Methods and findings

INTRO-HCV is a multicenter, randomized controlled clinical trial. Participants recruited from opioid agonist therapy (OAT) and community care clinics in Norway over 2017 to 2019 were randomly 1:1 assigned to the 2 treatment approaches. Integrated treatment was delivered by multidisciplinary teams at opioid agonist treatment clinics or community care centers (CCCs) for people with substance use disorders. This included on-site testing for HCV, liver fibrosis assessment, counseling, treatment, and posttreatment follow-up. Standard treatment was delivered in hospital outpatient clinics. Oral direct-acting antiviral (DAA) medications were administered in both arms. The study was not completely blinded. The primary outcomes were time-to-treatment initiation and sustained virologic response (SVR), defined as undetectable HCV RNA 12 weeks after treatment completion, analyzed with intention to treat, and presented as hazard ratio (HR) and odds ratio (OR) with 95% confidence intervals.

**Data Availability Statement:** All relevant data are within the manuscript and its Supporting information files.

**Funding:** The INTRO-HCV study was funded by The Norwegian Research Council (BEHANDLING, funding no 269855, grant received by LTF et al.) and the Western Norway Regional Health Authority (Helse Vest «Åpen prosjektstøtte», grant received by LTF et al.) with the Department of Addiction Medicine, Haukeland University Hospital as the responsible institution. The funders had no role in the study design, data collection and analysis, decision to publish, or preparation of the manuscript.

**Competing interests:** The authors LTF, CFA, JHV, SS, CO, FC, EML, KAJ, have declared that no competing interests exist. We have read the journal's policy and these authors of this manuscript have the following competing interests: AL has received lecture and advisory fees from Gilead, GSK and MSD. OJL has received project grants from AbbVie, Gilead and MSD. OD has received research grants from and been in advisory board for MSD, Abbvie and Gilead. PV has received an unrestricted research grant off Gilead. HM has received lecture and advisory fees from Gilead, Abbvie and MSD.

**Abbreviations:** CCC, community care center; DAA, direct-acting antiviral; HCV, hepatitis C virus; HIV, human immunodeficiency virus; HR, hazard ratio; IQR, interquartile range; ITT, intention-to-treat; OAT, opioid agonist therapy; OR, odds ratio; PWID, people who inject drugs; SVR, sustained virologic response.

Among 298 included participants, 150 were randomized to standard treatment, of which 116/150 (77%) initiated treatment, with 108/150 (72%) initiating within 1 year of referral. Among those 148 randomized to integrated care, 145/148 (98%) initiated treatment, with 141/148 (95%) initiating within 1 year of referral. The HR for the time to initiating treatment in the integrated arm was 2.2 (1.7 to 2.9) compared to standard treatment. SVR was confirmed in 123 (85% of initiated/83% of all) for integrated treatment compared to 96 (83% of initiated/64% of all) for the standard treatment (OR among treated: 1.5 [0.8 to 2.9], among all: 2.8 [1.6 to 4.8]). No severe adverse events were linked to the treatment.

## Conclusions

Integrated treatment for HCV in PWID was superior to standard treatment in terms of time-to-treatment initiation, and subsequently, more people achieved SVR. Among those who initiated treatment, the SVR rates were comparable. Scaling up of integrated treatment models could be an important tool for elimination of HCV.

## Trial registration

ClinicalTrials.gov.no NCT03155906

## Author summary

### Why was this study done?

- People who inject drugs (PWID) suffer from a high burden of disease including chronic hepatitis C virus (HCV) infection.

- Uptake of HCV treatment is often low in this population. Integrated care has been put forward as a more beneficial approach by incorporating the treatment of comorbid conditions into the treatment of substance use disorders, e.g., within specialized opioid agonist therapy (OAT) clinics.

- To date, there is limited evidence comparing standard HCV treatment with integrated treatment.

### What did the researchers do and find?

- In this randomized controlled trial, a 19 percentage points absolute increase in sustained virologic response (SVR) to HCV was seen among PWID who received integrated treatment compared to those who received standard treatment.

- The treatment initiation rates were 21 percentage points higher among those who received integrated treatment as compared to those who received standard treatment.

- The number needed to offer integrated treatment to achieve one additional SVR was 5.

**What do these findings mean?**

Integrated treatment programs for PWID with HCV infection could substantially improve overall treatment outcomes.

- Integrated treatment programs could be essential for HCV elimination and reducing morbidity and mortality among PWID.

## Introduction

The hepatitis C virus (HCV) epidemic has increased in many countries over the past few decades. In the United States alone, it is estimated that around 34,000 people died in 2017 due to severe hepatitis C complications such as cirrhosis and liver cancer [1]. HCV infection is endemic among people who currently inject or formerly injected drugs (PWID) [2]. Globally, 43% of the HCV disease burden is attributable to drug injection, and in high-income countries, this fraction is 79% [3]. In recent years, highly efficacious direct-acting antiviral (DAA) treatments for HCV infection have become available, prompting the World Health Organization (WHO) to develop a global health strategy to eliminate HCV infection as a public health threat by 2030 [4]. To achieve this goal, treatment models must address PWID [5].

Chronic HCV infection is a slowly progressive disease that can lead to liver cirrhosis and subsequent complications, including hepatocellular carcinoma and liver failure [6,7]. In a cohort of HCV-infected PWID, one-third developed advanced liver disease within 3 decades, and liver disease and drug overdose became equally important causes of death among individuals over 50 years [8,9]. Until recently, reports from opioid agonist therapy (OAT) clinics in Norway indicated that about half of patients receiving OAT had chronic HCV infection [10]. Furthermore, the prevalence of ongoing injecting drug use and the subsequent transmission of HCV is high in this population [11,12]. Thus, reaching PWID with HCV treatment is of critical importance for reducing both the HCV disease burden and HCV transmission.

Until 2014, HCV treatment was interferon based with moderate efficacy, considerable adverse effects, and an accordingly low treatment uptake, particularly among PWID [13]. The current availability of oral DAA has changed the HCV care paradigm and led to significant therapeutic optimism [12]. However, HCV treatment coverage for PWID is far from universal [14]. The World Hepatitis Summit in 2017 expressed concern over the lack of new healthcare models with a patient-centered service delivery and integrated collaborative approaches [15]. To reduce this inequity, new models of care must be developed. Models focusing on interdisciplinarity, availability, and accessibility, with decentralized clinics and frequent follow-ups, have been suggested [16–18]. Better evidence regarding these models of care for PWID is needed [19–21].

In western Norway, OAT is provided with supervised and observed intake of medications and frequent follow-ups by a multidisciplinary healthcare team that includes physicians, nurses, social workers, peer counselors, and psychologists [22]. Multidisciplinary care and support at community care centers (CCCs) are offered to PWID not receiving OAT. The multidisciplinary OAT clinics/CCC are well-suited delivery platforms to evaluate the efficacy of integrated HCV care for PWID. To date, given the low coverage of HCV treatment in many countries, surprisingly few trials with sufficient power have evaluated the efficacy of various treatment approaches for HCV among PWID [20].

This randomized controlled trial aims to compare the efficacy of integrated to standard HCV treatment for PWID.

## Methods

### Study design and setting

The INTRO-HCV study is a multicenter, randomized controlled trial. The target population was HCV-infected PWID eligible for treatment according to national guidelines. Patients with substance use disorders were treated at 1 of the 8 OAT clinics or 2 CCC in Bergen and Stavanger. For a more comprehensive description, a published study protocol is available [23].

OAT clinics located in each suburb served around 1,000 patients who were prescribed buprenorphine or methadone and, less commonly, other opioids or additional benzodiazepines or centrally acting stimulants [24]. This group of patients has a significant disease burden and limited ability to utilize standard healthcare. In the area surrounding the included clinics, around half of PWID had chronic HCV at the start of follow-up [25].

### Participants

The study participants were comprised of PWID (1) receiving follow-up from OAT clinics/ CCC from an included clinic involving follow-up at least once weekly; (2) who were diagnosed with detectable HCV RNA; (3) were eligible for treatment according to national HCV treatment guidelines (see S1 Text); and (4) who provided written informed consent before inclusion. Participants were excluded if they were currently receiving treatment for HCV, were coinfected with human immunodeficiency virus (HIV), or had severe extrahepatic HCV manifestations (e.g., cryoglobulinemia or membrano-proliferative glomerulonephritis), chronic kidney disease stage 4 to 5 (glomerular filtration rate $<30$ ml/min/1.73 m$^2$), or decompensated liver disease (Child–Pugh class B or C).

The study participants were PWID recruited from OAT clinics in Bergen ($n = 901$), Stavanger ($n = 130$), and CCC in Bergen ($n = 208$). Of these, 136/1,237 (11%) did not provide consent for participation. A total of 298 had untreated HCV with no exclusion criteria and were valid for randomization (Fig 1).

### Randomization and masking

Participants were randomized with a 1:1 ratio using blocks of 10 stratified by city and assigned into integrated ($n = 148$) or standard care ($n = 150$) for inclusion in the trial. Complete blinding was considered difficult and would have reduced external validity [26], although some masking measures were taken [23]. In short, randomization was disclosed to clinical staff providing treatment and follow-up. Patients were informed of key elements in the delivery of the respective intervention and follow-up they were randomized to, but not on information about the treatment and follow-up alternatives in the other arm, nor the hypotheses for the study.

### Interventions

Enrollment commenced in May 2017 and was completed in June 2019. For both arms, all the primary stages of the study—including recruitment, information, obtaining written consents, clinical interviews, completing the study surveys using appropriate instruments and blood sampling, and liver fibrosis assessment for diagnostic purposes—were performed by research nurses in close collaboration with the responsible multidisciplinary team at the OAT clinic/ CCC.

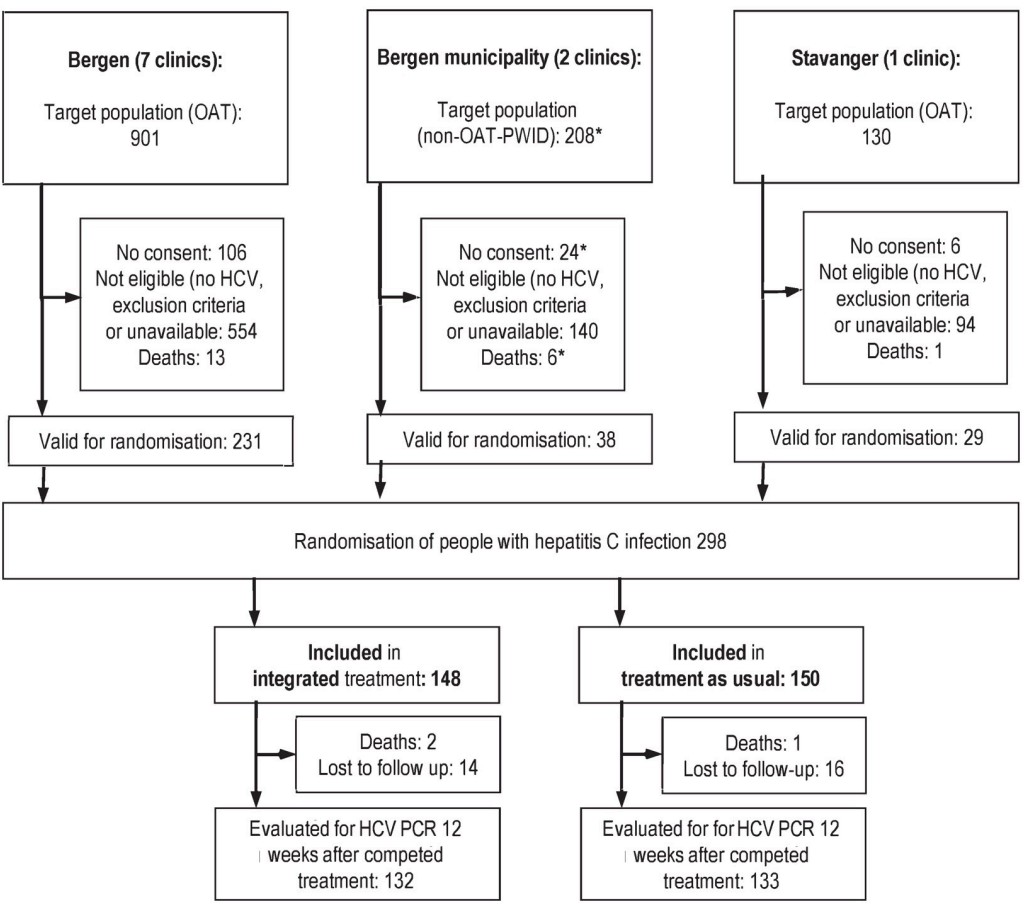

**Fig 1. Trial profile for the study.** *Estimated numbers. HCV, hepatitis C infection; OAT, opioid agonist therapy; PWID, people who inject drugs.

## Standard treatment

Participants eligible for standard treatment were referred for further assessment and treatment at the collaborating medical outpatient clinic located in a central hospital. An appointment was given and usually scheduled within a few weeks after the referral; the participants were informed of this by mail. The diagnostic assessment often involved additional blood samples and imaging before the initiation of DAA treatment. Once treatment was initiated, an electronic prescription was sent to a pharmacy during consultation. The DAA medications available were identical in the 2 trial arms and were prescribed in accordance with the national treatment guidelines. Norway has a tender-based system, and, during the period, elbasvir/grazoprevir, sofosbuvir/ledipasvir, or sofosbuvir/velpatasvir were the main combinations used (see S1 Text). Standard treatment participants were generally offered follow-up consultations every 4 weeks during treatment as well as a posttreatment assessment 12 weeks after completion. This typically involved a total of 4 to 5 consultation visits at the outpatient clinic.

For the standard treatment, patients needed to travel to the hospital clinic and pay for the transport themselves, a distance that ranged from 1 to 25 km. Patients in the standard treatment arm had standard follow-up in the OAT clinic for substance use disorders, and all other types of care—apart from HCV care—were integrated into the OAT follow-up. The OAT site staff encouraged participants to visit the infectious disease hospital clinics, but no further

extensive follow-up was given. There was a risk that scheduled appointments may overlap with other activities such as other healthcare, since arrangements were not coordinated. Blood samples were routinely taken every 4 weeks during treatment.

## Integrated treatment

Integrated treatment was delivered at OAT clinics or CCC by multidisciplinary teams in both of the settings. OAT clinics differed from CCC by offering OAT medications in addition to psychosocial approaches. Psychosocial approaches were also provided at CCC. The multidisciplinary team at the OAT clinics were equipped with specialists in addiction medicine who were responsible for the OAT and other medical follow-ups and also psychologists providing mental health treatment. In both settings, nurses and social workers in cooperation with peer counselors provided most of the daily follow-ups of the patients. All these professionals were existing clinical staff who closely worked together with the research nurses in management of the interventions and evaluations during the study period. For those found to be eligible for HCV treatment, DAA medications were administered by a nurse at the OAT clinic/CCC after prescription from a specialist physician. All HCV treatment and scheduled follow-up during treatment were given in parallel with the delivery of OAT medications and other care in line with the study protocol (usually several times per week). Dispensation was therefore adapted to the level of functioning for each individual patient. For the most severely ill patients with the lowest level of daily functioning and high intake of multiple drugs, OAT medications and HCV treatment were normally dispensed daily in the OAT clinic, and intake was observed by a nurse. For patients with a high level of functioning, dispensation could be done, for example, on a weekly basis.

Frequency of blood sampling during integrated treatment was mainly taken according to clinical indication (e.g., suspected adverse effects or complications). Less frequent blood sampling may reduce discomfort compared to standard treatment, where blood samples are usually taken every 4 weeks during treatment. To assess treatment efficacy, a new blood sample was drawn by a research nurse 12 weeks after treatment completion.

## Outcomes and assessments

The primary outcome measures were as follows:

- Treatment uptake, defined as initiation of DAA among participants eligible for HCV treatment, was assessed with time-to-event analyses and initiation status. In the time-to-event analyses, the first time point was when participants were diagnosed with chronic HCV infection, and the second time point was when treatment was initiated. We also present stratified analyses on patient-related delay (including time since missed appointments) and system-related delay (including delay until received appointment).

- Sustained virologic response (SVR), defined as undetectable HCV RNA 12 weeks after completion of treatment with DAA. This was assessed as a dichotomous SVR status. The virologic blood samples in Bergen were analyzed at the Department for Microbiology at Haukeland University Hospital (accredited by ISO standard 15189) after being centrifuged at each study clinic before transfer. Similar procedures were used in Stavanger.

Transient elastography was used in the screening of cirrhosis (median of 10 measurements >12.5 kPa on an empty stomach using FibroScan 430 mini) [27,28], and hepatic encephalopathy was assessed according to the West Haven criteria [29]. For details on clinical and demographic data, see Table 1.

**Table 1. Demographic and clinical characteristics at baseline for participants in standard and integrated treatment groups presented with numbers in each group with percentages or median (25%–75%) for age and body mass index.**

|  | Standard treatment | Integrated treatment |
|---|---|---|
| *n* (%) | **150 (50%)** | **148 (50%)** |
| Age | 42 (34–50) | 44 (36–52) |
| Education: <10 years | 82 (57%) | 77 (52%) |
| Education: 10–12 years | 47 (33%) | 57 (39%) |
| Education: >12 years | 14 (10%) | 14 (9%) |
| Male | 121 (81%) | 108 (73%) |
| Homelessness | 19 (13%) | 22 (15%) |
| Social security benefits as income | 131 (87%) | 146 (99%) |
| Formal work as income | 13 (9%) | 3 (2%) |
| OAT |  |  |
| Buprenorphine based | 69 (46%) | 72 (49%) |
| Methadone | 56 (37%) | 57 (39%) |
| Non-OAT (receiving CCC) | 23 (15%) | 18 (12%) |
| Injecting drug use (ever) | 133 (100) | 147 (100) |
| Injecting drug use (last 6 months) | 84 (63) | 84 (57) |
| Injecting drug use (last 30 days) | 62 (47) | 62 (42) |
| Frequency of follow up (days/week) | 5 (3–6) | 5 (3–6) |
| Substance use last 30 days |  |  |
| Illicit opioids | 38 (29) | 33 (22) |
| Amphetamines or cocaine | 63 (47) | 68 (46) |
| Benzodiazepines | 78 (59) | 86 (59) |
| Cannabinoids | 88 (66) | 111 (76) |
| Tobacco | 123 (92) | 138 (94) |
| Alcohol | 78 (59) | 81 (55) |
| Body mass index (kg/m$^2$) | 24 (22–28) | 24 (22–28) |
| **Genotype** |  |  |
| Genotype 1 | 50 (34%) | 51 (35%) |
| Genotype 2 | 6 (4%) | 2 (1%) |
| Genotype 3 | 89 (61%) | 94 (64%) |
| Other genotypes | 2 (1%) | 0 (0%) |
| Probable fibrosis* (not cirrhosis) | 28 (21%) | 34 (25%) |
| Probable cirrhosis** | 14 (11%) | 21 (15%) |

Missing values are excluded from percentages.

*Probable fibrosis (not cirrhosis): elastography measure of 7.0–12.5 kPa.

**Probable cirrhosis: elastography measure of >12.5 kPa.

CCC, community care center; OAT, opioid agonist therapy.

The blood samples for the primary outcome measures were collected at the OAT clinics/ CCC for participants randomized into integrated care and at the hospital outpatient clinic for the participants randomized into standard treatment. However, for participants who did not come to the assessment at the hospital clinic 12 weeks after treatment, blood samples were collected at the OAT clinics/CCC.

## Statistical analyses

A sample size was calculated with the following assumptions: 90% power with a two-sided alpha ($\alpha$) error of 5%, 50%/80% rates of SVR among all randomized to standard/integrated

treatment, up to 33% lost to follow-up, and equal proportions between the groups, with 87 required per group and 174 in total. We also calculated sample size requirements to detect a 1.5 hazard ratio (HR) difference in time to treatment initiation between the arms (post hoc), with 90% power and an alpha of 5%, yielding 134 required per group and 268 in total. Due to logistical reasons such as study recruitment being very rapid during the last month, the number of participants exceeded the precalculated necessary sample size.

Analysis was conducted in Stata SE16 and followed the CONSORT guidelines [30]. We present intention-to-treat (ITT) analyses by including all participants based on randomization, where those lost to follow-up are assumed to persist as HCV infected. Per protocol analyses are also presented, where those with protocol violations are excluded. For SVR per protocol analyses, also those without valid HCV PCR measures 12 weeks after completed treatment are excluded. All tests are two-sided. Descriptive results and efficacy estimates are presented with 95% confidence intervals. The statistical significance is set at $p < 0.05$. Categorical variables are summarized as percentages and continuous variables as medians with interquartile ranges (IQRs). The time-to-event analyses are presented with Kaplan–Meier plots for both treatment arms and Cox regression using randomization arm as independent variable and time-to-treatment initiation as dependent variables, presented as HRs with 95% confidence intervals. For binary outcomes of SVR, logistic regression analyses are presented as odds ratios (ORs) with 95% confidence intervals. The proportional hazards assumption was also checked using Schoenfeld residuals. The ITT analyses did not violate the assumption significantly, but some violation was seen with the per protocol analyses (S1 Table). Thus, sensitivity analyses with split time for treatment initiation was conducted with time splits at 2, 6, and 12 months. To account for potential within clinic variation, we also did sensitivity analyses adjusting for clustering in the regression models. Additional post hoc subgroups analyses stratified for gender, geographic location, age group, living condition, likely fibrosis and cirrhosis, and injecting drug behavior during the last 6 months are presented, as are analyses taking potential clustering effects into consideration. Tests for the interactions in the models are presented with $p$-values for the interaction between the subgroup and arm when including the arm and each subgroup separately. We also present subgroup Kaplan–Meier plots for the dominant genotypes. All severe adverse effects are reported.

## Ethics

The study has been approved by regional ethical committee (no. 2017/51/REK Vest). The trial was conducted in accordance with the Declaration of Helsinki and other international conventions and with Good Clinical Practice and Good Laboratory Practice standards [31,32]. Written informed consent was obtained from each participant.

## Results

The characteristics of study participants were similar in the 2 arms with a median age of 44 years (IQR 36 to 52) in the integrated care arm and 42 years (IQR 34 to 50) in standard treatment arm (Table 1). The majority for both arms were male (77%, 229/298), had received buprenorphine-based OAT (47%, 141/298) or methadone (38%, 113/298), had injected drugs within the last 6 months (60%, 168/280), and received supervised and observed treatment with a median of 5 times per week (IQR 3 to 6) in an OAT outpatient clinic. Homelessness, defined as not having a rented or owned home nor living together with family, was reported among 14% (41/298) of study participants. All participants had a history of injecting drug use. During the last month, 88% (261/298) had smoked tobacco and 67% (199/298) cannabinoids, 44% (131/298) used stimulants such as amphetamine, methamphetamine, or cocaine, 55% (164/

**Table 2. Primary end point analysis presented for treatment initiation with both ITT and PP analyses.**

| Outcome | Events, No. (%) | | Absolute increase (%, 95% CI) | Cox/glm.reg. HR/OR | Cox/glm.reg. (clustering) |
|---|---|---|---|---|---|
| | **Integrated** | **Standard** | | | |
| Treatment initiation, ITT | 145 (98) | 116 (77) | 21 (14–28) | 2.2 (1.7–2.9) | 2.2 (1.6–3.0) |
| Treatment initiation, PP | 145 (98) | 107 (82) | 16 (9–22) | 1.9 (1.5–2.5) | 1.9 (1.5–2.5) |
| HCV SVR (initiated), ITT | 123 (85) | 91 (78) | 6 (−3–16) | 1.5 (0.8–2.9) | 1.5 (0.9–2.6) |
| HCV SVR (init. + test), PP | 123 (94) | 84 (88) | 7 (0–14) | 2.5 (0.9–6.6) | 2.5 (1.2–5.2) |
| HCV SVR (all), ITT | 123 (83) | 96 (64) | 19 (9–29) | 2.8 (1.6–4.8) | 2.8 (1.7–4.5) |
| HCV SVR (all + test), PP | 123 (93) | 84 (73) | 20 (11–29) | 5.0 (2.3–11) | 5.0 (2.3–11) |
| SAE (assumed linked) | 0 (0) | 0 (0) | | | |

SVRs are presented for those who initiated treatment (marked "initiated") and for all who were randomized (marked "all"). No related SAEs were observed. Both number of events, absolute increase, and relative changes assessed with HR for treatment initiation (analyzed with Cox regression) and OR for SVR (analyzed with GLM regression).

GLM, generalized linear model; HCV, hepatitis C virus; HR, hazard ratio; ITT, intention-to-treat; OR, odds ratio; PP, per protocol; SAE, severe adverse event; SVR, sustained virologic response.

298) had used benzodiazepines, 53% (159/298) had used alcohol, and 24% (71/298) used illicit opioids such as heroin. For 23% (62/270) participants, the elastography indicated fibrosis but not cirrhosis, and for 13% (35/270), cirrhosis was likely. Almost all had HCV infection with genotype 1 or 3 (97%, 284/294).

In the integrated care arm, 98% (145/148) initiated treatment, and 94% (139/148) started within 1 year of referral (Table 2). Among people randomized to standard treatment, 77% (116/150) initiated treatment, and 72% (108/150) started within 1 year of referral. Median time to HCV treatment initiation was 71 days (IQR 39 to 112) in the integrated care arm, of which system delay (until receiving time for appointment) contributed to 51 days (IQR 23 to 104). In contrast, for the standard treatment arm, the median time to HCV treatment initiation was 120 days (IQR 47 to 248), with a system delay of 81 days (IQR 32 to 192; Fig 2). The HR for the time to initiating treatment in the integrated arm was 2.2 (1.7 to 2.9) compared to standard treatment. For the binary treatment initiation outcome, logistic regression indicated an OR of 6.8 (3.1 to 15) for initiating treatment within 1 year in the integrated arm compared to the standard arm. For per protocol analyses excluding participants with protocol violations, the HR for the time to initiating treatment was 1.9 (1.5 to 2.5) for integrated compared to standard treatment (S1 Fig).

Among all participants who were randomized, the ITT analyses showed that the SVR was achieved in 123 (83%) in the integrated treatment arm and 96 (64%) in the standard treatment arm. For patients who initiated treatment, SVR was achieved in 85% receiving integrated treatment compared to 83% receiving standard treatment. In the per protocol analyses when excluding persons with protocol violations or no valid posttreatment PCR test, SVR was confirmed among 94% (123/130) receiving integrated treatment and 88% (84/96) receiving standard treatment, OR 2.5 (0.9 to 6.6).

Subgroup analyses of the SVR among those who initiated treatment (Fig 3) indicated that the intervention increased SVR most among participants above 50 years of age but also tended to increase SVR more among female participants and those with injecting drug use not receiving OAT. Most groups had superior outcomes from integrated treatment compared to standard treatment when comparing SVR among all who were randomized (Fig 4 and S2 Fig).

For the sensitivity analyses with split time (S1 Table), the HR for time to treatment initiation comparing the integrated arm with the standard arm in the time range 0 to 2 months was

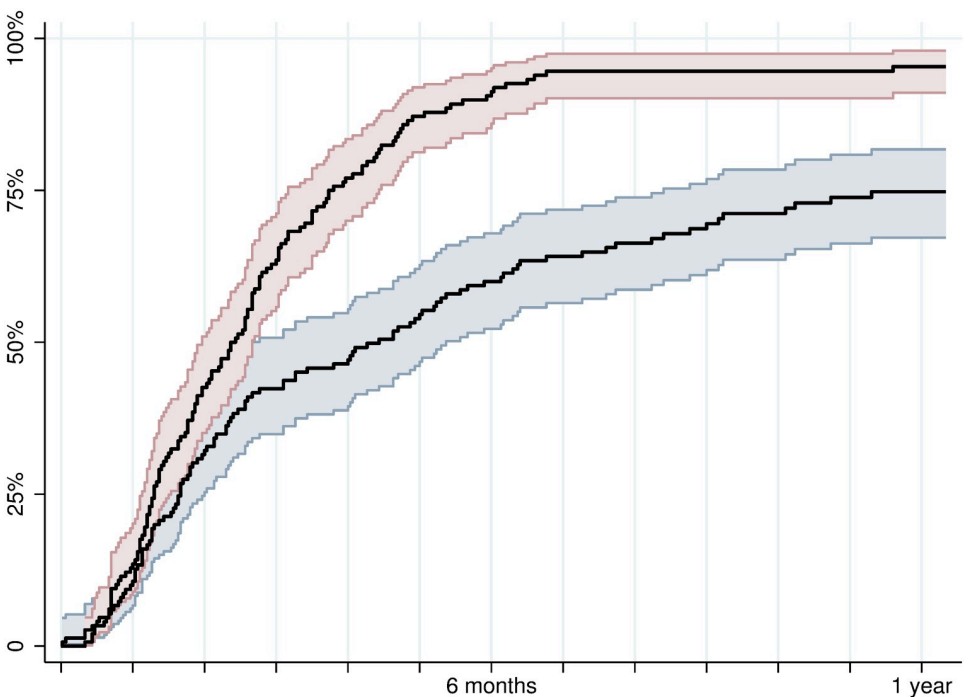

**Fig 2. Time to hepatitis C treatment initiation presented with Kaplan–Meier plot.** Red line/area indicates proportion of participants in integrated arm initiation treatment with confidence intervals. Blue dashed line/area indicates proportion of participants in standard treatment arm initiation treatment.

1.4 (1.0 to 2.1), between 2 and 6 months was 3.5 (2.4 to 5.1), and between 6 and 12 months was 2.2 (1.0 to 4.8). There were no clear differences in time-to-treatment initiation between the genotypes (S3 and S4 Figs). Out of 101 persons with genotype 1, 55% were treated with elbasvir/grazoprevir and 40% with sofosbuvir/ledipasvir. Among 183 persons with genotype 3, 93% were treated with sofosbuvir/velpatasvir. Except for the sofosbuvir/ledipasvir combination that was usually given for 8 weeks, other treatments were generally administered for 12 weeks.

Three deaths were recorded among the trial participants, 2 receiving the integrated treatment and 1 receiving standard treatment. These deaths were linked to overdose, suicide, and murder, with none being associated with the HCV treatment. In addition, there was a near fatal suicide attempt concerning an individual undergoing standard treatment, also considered unlikely to be linked to the treatment. No other serious adverse events were recorded. One-tenth of the participants had noteworthy side effects such as fatigue, nausea, headache, and stomach pain during the treatment. Approximately half of these ceased the treatment due to the side effects. For a few participants in the standard treatment, there was an issue with medications being lost due to theft.

## Discussion

This study provides strong evidence supporting the effect of integrating HCV treatment among PWID into OAT/CCC. The trial showed a substantial increase in SVR among those eligible for treatment, with a 19 percentage points absolute difference seen in favor of the integrated treatment model compared to standard treatment. For treatment initiation, a 21 percentage points absolute difference was seen in favor of the integrated treatment model and a reduction by half in the time to treatment initiation. No associated serious adverse events were recorded.

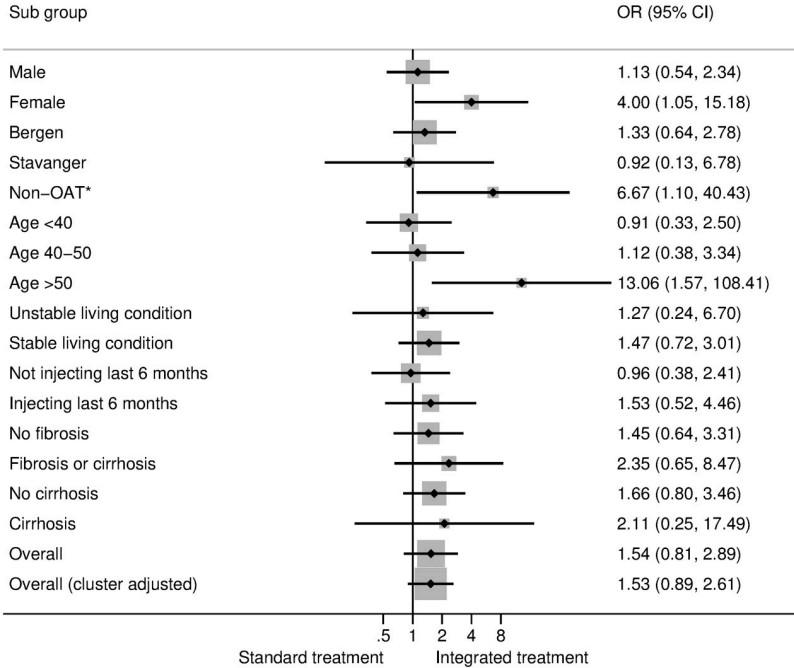

**Fig 3. Subgroup analyses for binomial logit regression of SVR of hepatitis C for those who initiated treatment comparing the integrated arm with the standard arm (post hoc ITT analyses).** Subgroup effects are presented for gender, the type of treatment center, age group, living condition, injecting drug use behavior the last 6 months, fibrosis and cirrhosis, and overall effects (without and with cluster adjustment)**. *Non-OAT: patients who did not receive OAT. **Interaction tests: sex: $p = 0.069$, site: $p = 0.737$, OAT/non-OAT: $p = 0.095$, age: $p = 0.023$, living condition: $p = 0.367$, injecting drugs: $p = 0.309$, fibrosis/cirrhosis: $p = 0.605$. ITT, intention-to-treat; OAT, opioid agonist therapy; OR, odds ratio; SVR, sustained virologic response.

Integrated treatment models have shown they can provide an excellent opportunity to improve treatment initiation/uptake to HCV treatment, and so could be essential to the aim of providing universal coverage of HCV among marginalized populations with substance use disorders [15]. Representing a model of care suited to engage OAT patients in HCV care, this study should inform HCV elimination efforts internationally and replace the current referral-based standard treatment for this patient group.

A previous three-arm randomized controlled trial (PREVAIL) of 3 models of care for the treatment of HCV among PWID showed that directly observed treatment was more effective in improving adherence to HCV treatment than self-administered individual treatment [33]. However, this study from the Bronx in New York City was conducted during a transition time when interferon was still in use. There was no comparison with the current standard treatment, and it was not powered to assess differences in SVR between the treatment models. Our study is among the first trials in the DAA era to examine the efficacy of integrated treatment compared to referral-based standard treatment. HCV treatment has become increasingly simple and effective to administer, except for a small minority of patients with severe liver disease, and so tertiary healthcare is less necessary. We hypothesize a significant effect of integrated treatment and care models in the management and follow-up of a range of comorbid physical and mental conditions among people currently injecting substances. These populations might be more vulnerable and have more difficulties in daily functioning, making it challenging to utilize usual healthcare services. There is an urgent need to assimilate the health system and care services to improve access to highly efficacious HCV treatment for the most vulnerable population groups.

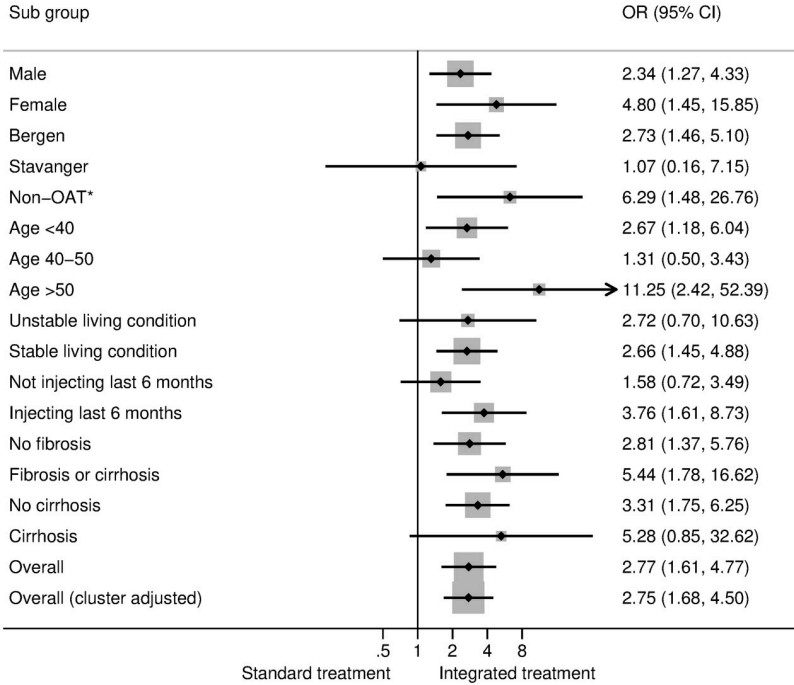

**Fig 4. Subgroup analyses for binomial logit regression of SVR of hepatitis C for all who were randomized comparing the integrated arm with the standard arm (post hoc ITT analyses).** Subgroup effects are presented for gender, the type of treatment center, age group, living condition, injecting drug use behavior the last 6 months, fibrosis and cirrhosis, and overall effects (without and with cluster adjustment)[**]. *Non-OAT: patients who did not receive OAT. [**]Interaction tests: sex: $p = 0.231$, site: $p = 0.358$, OAT/non-OAT: $p = 0.237$, age: $p = 0.041$, living condition: $p = 0.928$, injecting drugs: $p = 0.972$, fibrosis/cirrhosis: $p = 0.382$. ITT, intention-to-treat; OAT, opioid agonist therapy; OR, odds ratio; SVR, sustained virologic response.

In our study, only 3 out of 4 patients referred to standard treatment initiated treatment. This is higher than findings from other, such as among PWID in Melbourne, Australia, where only 1 in 3 of those referred to hospital actually commenced HCV treatment [34]. It is worth noting that participants in the standard treatment arm in our trial received an integrated assessment beyond what was common in many similar settings in Norway and probably also in other countries. The difference between our trial and the study from Australia might be related to our standard arms receiving these integrated assessments including blood samples for genotyping as well as elastography, which reduced the number of visits in standard treatment before treatment initiation. Among those who initiated treatment across in our trial and had valid SVR tests, 94% in the integrated arm and 88% in the standard arm achieved SVR, which is comparable to a trial on DAA medications where 92% achieved SVR [35]. Studies on integrated diagnostics have already shown improved case findings [36]. Thus, the difference between standard treatment without active case finding procedures and our integrated assessment and treatment model may even have been larger than the differences seen in our trial.

Our trial involves several strengths and limitations. The treatment evaluation (assessment of outcomes) and treatment follow-up were undertaken by independent teams. The study is individually randomized with balanced groups, minimizing the potential for confounding. The study size was also more than sufficient to answer the primary objectives with high precision. Moreover, the study was also funded by public sources minimizing potential conflicts of

interest. In terms of safety, the frequent follow-up of the participants likely improved detection of potential adverse effects of applied medications. The population of PWID includes a large proportion that is struggling with standard treatment pathways due to their substance use disorders. For individuals with higher levels of functioning and less need for regular treatment follow-up, the need for treatment integration might be less important and have less beneficial effect.

It could be argued that integrated care provides added benefits in contrast to hospital care, related to culture, attitudes, and philosophy of care in the direction of anti-stigma, psychological safety, and relational continuity. Integrated care may also provide additional support, engagement, and knowledge of substance use via specialized multidisciplinary teams. A limitation of the present study is that the benefits of the separate elements of the integrated care model cannot be differentiated. Still, we argue that the benefits of integrated care are related to a combination of these benefits in addition to ease of access. This is a first step in developing a better care model, with the next step being to disentangle the effects of the different components of the integrated care model.

It should be noted that OAT in the region of study was performed and administered at centralized clinics. In regions where OAT is provided by primary care, models of HCV care might need local adaptations. In the OAT population, 11% had received HCV treatment prior to the trial, and a similar percentage did not provide consent to study participation. Thus, some participants that may be less hard to reach were treated outside this trial. As 5 times as many of the target population were treated inside the trial as before study initiation, this is unlikely to cause a substantial selection bias. When taking potential clustering within the 10 treatment centers into consideration, only minor changes in the confidence intervals were observed. We adhered to the commonly used standard of SVR 12 weeks after treatment completion. An earlier outcome measure might have reduced loss to follow-up and improved our evaluation of who had been successfully treated. However, most of the loss to follow-up before SVR testing was among persons not initiating treatment, and so not achieving SVR is likely to be a robust assumption. In the time prior to the national guidelines that allowed for immediate HCV treatment, there was some delay in treatment initiation for several participants in both arms of the study. This might have negatively impacted their treatment outcomes, although it was balanced between the arms. For the trial, complete blinding was not possible. However, due to the assessment of a biological primary outcome such as HCV SVR, we assume that information bias is unlikely to have substantially distorted our findings. One can view trials as a continuity from efficacy trials optimizing internal validity through homogeneous population to effectiveness trials optimizing external validity with a more heterogeneous population. This trial is more toward the effectiveness side of the spectrum.

An important question for trials of different treatment models is how these models can be successfully sustained outside the trial settings. As existing clinic staff were used to deliver the intervention within existing clinical frameworks and settings, the intervention is considered sustainable.

In conclusion, integrating hepatitis C treatment into multidisciplinary OAT clinics and CCCs for PWID was found to be superior to standard treatment at hospital, both in terms of time-to-treatment initiation and subsequently achieving SVR among more patients. Among those who initiated treatment, the rates of SVR were comparable. No differences in severe adverse effects were shown between the arms. Scaling up of integrated treatment models could be an essential tool for the elimination of hepatitis C infection among people with substance use disorders.

## Supporting information

**S1 Text. Text file including information on Norwegian national HCV treatment guidelines during the study period and description of shared data file.** HCV, hepatitis C virus.
(PDF)

**S1 Table. Testing of proportional hazards assumption in Cox regression for effect of integrated vs standard treatment with SRp and sensitivity analyses with split time for treatment initiation with ITT and PP analyses.** ITT, intention-to-treat; PP, per protocol; SRp, Schoenfeld residuals *p*-values.
(PDF)

**S1 Fig. Time to hepatitis C treatment initiation in PP analyses taking change in treatment arm into consideration and excluding people with unconfirmed results.** PP, per protocol.
(PDF)

**S2 Fig. Subgroup analyses for binomial logit regression of SVR of hepatitis C for those who initiated treatment (post hoc PP analyses).** Subgroup effects are presented for gender, the type of treatment center, age group, living condition, injecting drug use behavior the last 6 months, fibrosis and cirrhosis, and overall effects (without and with cluster adjustment). PP, per protocol; SVR, sustained virologic response.
(PDF)

**S3 Fig. Time to hepatitis C treatment initiation for genotype 1.**
(PDF)

**S4 Fig. Time to hepatitis C treatment initiation for genotype 3.**
(PDF)

**S1 CONSORT checklist. CONSORT checklist for reporting of the trial.**
(PDF)

**S1 Data. Data sharing file.**
(CSV)

## Acknowledgments

We thank the INTRO-HCV Study Group (see list below) and devoted clinical staff for their enthusiasm during the planning of the project. We also thank Nina Elisabeth Eltvik and Christer Kleppe for valuable help and input during the planning and preparation phases.

   **The INTRO-HCV Study Group participating investigators are the following:** Bergen: Christer Frode Aas, Vibeke Bråthen Buljovcic, Fatemeh Chalabianloo, Jan Tore Daltveit, Silvia Eiken Alpers, Lars T. Fadnes (principal investigator), Trude Fondenes Eriksen, Rolf Gjestad, Per Gundersen, Velinda Hille, Kristin Holmelid Håberg, Kjell Arne Johansson, Rafael Alexander Leiva, Siv-Elin Leirvåg Carlsen, Martine Lepsøy Bonnier, Lennart Lorås, Else-Marie Løberg, Mette Hegland Nordbotn, Maria Olsvold, Christian Ohldieck, Lillian Sivertsen, Hugo Torjussen, Jørn-Henrik Vold, and Jan-Magnus Økland; Stavanger: Tone Lise Eielsen, Nancy Laura Ortega Maldonado, and Ewa Joanna Wilk; proLAR: Ronny Bjørnestad, Ole Jørgen Lygren, and Marianne Cook Pierron; Oslo: Olav Dalgard, Håvard Midgard, and Svetlana Skurtveit; Bristol: Aaron G. Lim and Peter Vickerman (in alphabetical order of surname).

## Author Contributions

**Conceptualization:** Lars T. Fadnes, Rafael Alexander Leiva, Christian Ohldieck, Svetlana Skurtveit, Olav Dalgård, Else-Marie Løberg, Kjell Arne Johansson.

**Data curation:** Lars T. Fadnes.

**Formal analysis:** Lars T. Fadnes, Jørn Henrik Vold.

**Funding acquisition:** Lars T. Fadnes, Else-Marie Løberg, Kjell Arne Johansson.

**Investigation:** Lars T. Fadnes, Christer Frode Aas, Jørn Henrik Vold, Rafael Alexander Leiva, Christian Ohldieck, Fatemeh Chalabianloo, Ole Jørgen Lygren, Kjell Arne Johansson.

**Methodology:** Lars T. Fadnes, Christian Ohldieck, Svetlana Skurtveit, Olav Dalgård, Peter Vickerman, Else-Marie Løberg, Kjell Arne Johansson.

**Project administration:** Lars T. Fadnes, Christian Ohldieck, Else-Marie Løberg.

**Resources:** Lars T. Fadnes, Christian Ohldieck, Else-Marie Løberg.

**Software:** Lars T. Fadnes.

**Supervision:** Lars T. Fadnes, Svetlana Skurtveit, Ole Jørgen Lygren, Olav Dalgård, Peter Vickerman, Håvard Midgard, Else-Marie Løberg, Kjell Arne Johansson.

**Validation:** Lars T. Fadnes, Jørn Henrik Vold, Rafael Alexander Leiva, Kjell Arne Johansson.

**Visualization:** Lars T. Fadnes.

**Writing – original draft:** Lars T. Fadnes.

**Writing – review & editing:** Lars T. Fadnes, Christer Frode Aas, Jørn Henrik Vold, Rafael Alexander Leiva, Christian Ohldieck, Fatemeh Chalabianloo, Svetlana Skurtveit, Ole Jørgen Lygren, Olav Dalgård, Peter Vickerman, Håvard Midgard, Else-Marie Løberg, Kjell Arne Johansson.

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
