## [Editor Report · Decision Letter 0]

3 Sep 2020

Dear Dr Fadnes, 

Thank you for submitting your manuscript entitled "Integrated treatment of hepatitis C virus infection among people who inject drugs: A multi-center randomized controlled trial (INTRO-HCV)" for consideration by PLOS Medicine.

Your manuscript has now been evaluated by the PLOS Medicine editorial staff and I am writing to let you know that we would like to send your submission out for external peer review.

Kind regards,

Helen Howard, for Clare Stone PhD 

Acting Editor-in-Chief

PLOS Medicine 

plosmedicine.org

---

## [Decision Letter · Decision Letter 1]

12 Oct 2020

Dear Dr. Fadnes,

Thank you very much for submitting your manuscript "Integrated treatment of hepatitis C virus infection among people who inject drugs: A multi-center randomized controlled trial (INTRO-HCV)" (PMEDICINE-D-20-04289R1) for consideration at PLOS Medicine. 

[LINK]

In light of these reviews, I am afraid that we will not be able to accept the manuscript for publication in the journal in its current form, but we would like to consider a revised version that addresses the reviewers' and editors' comments. Obviously we cannot make any decision about publication until we have seen the revised manuscript and your response, and we plan to seek re-review by one or more of the reviewers. 

We expect to receive your revised manuscript by Nov 02 2020 11:59PM. Please email us (plosmedicine@plos.org) if you have any questions or concerns.

We look forward to receiving your revised manuscript. 

Sincerely,

Emma Veitch, PhD

PLOS Medicine

On behalf of Artur Arikainen, PhD, Associate Editor, 

PLOS Medicine

plosmedicine.org

*We'd recommend adding some context to the Abstract regarding the study setting, ie where participants were recruited (this info is in the full manuscript Methods but would be good to add to Abstract too) - eg, participants recruited from opioid agonist therapy clinics in Norway.

*We'd also suggest adding info to the Abstract on the randomization ratio (eg 1:1 to study arms) as this info is very brief to add but otherwise readers are not clear on the design.

*In the last sentence of the Abstract Methods and Findings section, please add a brief note about any key limitation(s) of the study's methodology.

*Please restyle the author summary section so that each subsection has bulleted sentences (eg see this paper for an example) - https://journals.plos.org/plosmedicine/article?id=10.1371/journal.pmed.1003358

Comments from the reviewers:

Reviewer #1: This paper describes a large and well conducted RCT comparing integrated care with standard care for treatment of HCV. It is the largest study of its kind published in the OAT setting. The findings are of considerable interest and the analysis is relevant and clear with the paper will presented. One of the strengths of the study is that it was performed in a real-world clinical setting with a population who have been recently using drugs, on OAT, and with other concurrent problems including homelessness (how defined?).

Fibroscan data are referenced with the implication these are included in Table 1, but I could not see them there. It would be worth noting the proportion with elevated scores (likely to be 25% or more) and the effectiveness of this model in that sub-population. If these were referred to the specialist clinic, did they attend, or was it still better to treat those with relatively advanced fibrosis in the OAT clinic setting?

One concern I have about this type of project is that the potential for translation into usual case is far from clear. It would be worth some discussion of this point: the intervention was undertaken by research staff and not by existing clinic staff. How can this effective model of care be successfully sustained after the end of the trial period?

Table 2 presents the effect size as "reduction" whereas it makes more sense to describe the difference as an 'increase'.

Fig 3: asterisk next to 'non-OAT' is missing

Reviewer #2: Alex McConnachie, Statistical Review

Fadnes et al present the results of a trial of integrating treatment of hepatitis C infection into existing clinic care amongst injecting drug users. This review considers the use of statistics in the paper.

Whilst the results of this study are important, in some ways they are not that surprising, since they show that incorporating an efficacious treatment into standard care in an integrated way, is more effective than incorporating it by referral to another service, which required additional time and money on the part of the patient. Nevertheless, this is important to demonstrate.

There are two main analyses. Time to start of treatment is analysed using Cox regression. The authors should check the proportional hazards assumption, but otherwise this is well presented, and is an appropriate method of analysis.

The second main analysis is of sustained virologic response, 12 weeks after completion of treatment. This is also analysed as a time-to-event variable, using Cox regression, though in this case I feel it is less appropriate. In survival analysis, the model assumes that all those who have not had the event are at risk of having it, up until they have the event or they reach the end of follow-up. I am not sure that is the case here. Only those who start treatment can achieve SVR, and they either achieve it or they do not when assessed 12 weeks after completion of treatment. The data can be analysed using Cox regression, but it seems conceptually incorrect; the main factor driving the shorter time to SVR with the intervention is simply the time to treatment. My interpretation of the data is that the intervention reduces the time to treatment, without affecting the efficacy of the treatment in those who receive it.

That is how I would present these data: treatment engagement as a time-to-event variable (as done already), but SVR as a binary outcome. Given that the intervention reduces the time to treatment, one aim of the SVR analysis is to show that the intervention is not associated with reduced efficacy in those who receive it. To me, that tells a more coherent story.

SVR can also be looked at within the whole randomised population, but I would again treat it as a binary variable, within a fixed time frame (e.g. 12 months). This will show that the combined effect of the intervention, i.e. reducing time to treatment without affecting treatment efficacy, results in improved SVR rates overall at 12 months.

Many of the components of this presentation are in the paper already, so all it needs is a slight shift in emphasis.

There were a few other minor things that I noticed.

In the abstract, at the start of the second paragraph under "Methods and findings", I think the number randomised to standard treatment is missing. In the same paragraph, the percentage achieving SVR is reported, but without a time frame; is it 12 months?

In "What did the researchers do and find?" the number needed to treat is reported, though again, is this the right concept? The intervention is not a "treatment", but the offer of treatment within an existing care programme. Is there such a thing as the number needed to offer?

The sample size section is incomplete for my liking; there is not enough information to replicate the calculation. Was the figure of 99 per group based on a time to event analysis, or a binary outcome analysis? Either way, we need to know what the control group event rate is, and whether a 25% increase is an absolute difference, or a relative effect. I am not clear on the need to allow for clustering effects, given that the randomisation was not at the site level, though the fact that the intervention was delivered at the site level, there could be a case for it. However, the current text does not make it clear what assumptions were made in making this adjustment. I was not convinced by the need to allow for subgroup analyses being a reason to over-recruit; I would have thought that a much greater inflation would be needed for this.

Defining a Per Protocol analysis in terms of analysing according to treatment given is not right - those who received the wrong intervention should be considered protocol deviations, and would be excluded from a per protocol analysis. A better term might be "As Treated".

When describing the logistic regression analysis of the binary outcomes of treatment initiation and SVR, the authors should taken more care to specify the time point at which these are assessed.

The subgroup analyses should fit interaction terms within the regression models to assess whether any differences in intervention effects between subgroups are greater than might be expected by chance. It is not enough to report the intervention effect estimates in each subgroup and draw conclusions. In reference to Figure 3, the paper currently claims stronger effects for men than for women, even though the HR reported for women is larger than for men; however, given the total overlap between the confidence intervals for men and women, I doubt very much that there is any evidence of a difference in intervention effects between the two groups.

In the second paragraph on page 8, I do not think it makes sense to report the percentage starting treatment within 12 months out of those who start at any time; it would be better to give the percentage of all participants. When reporting the crude difference between groups, the chi-square statistic is not of interest; better to give the estimated difference, or odds ratio, between groups, with a 95% confidence interval. In the "per protocol" analysis of treatment initiation, those with unknown SVR after treatment are excluded (if I am reading it correctly). I am not sure this makes sense, as the outcome for this analysis is whether treatment was started, which is unaffected by whether or not SVR after 12 weeks is known or not.

Analysis by genotype is presented in the results without being mentioned in the methods section.

Though mentioned in the methods section, analysis allowing for clustering is not mentioned in the results.

In Figure 1, it is reported that 132 people in the standard care arm were evaluated for SVR, but it is not clear how this is possible, if only 116 initiated treatment.

Reviewer #3: This is a well designed and conceptualised RCT of integrated HCV treatment by multidisciplinary teams in opioid treatment settings, which will make a worthwhile contribution to the literature. The statistical approaches are sound.

The mention of COVID-19 in the introduction is interesting but requires more in-depth consideration to be meaningful. In particular, the statement “This could even become essential if, for example a vaccine for COVID-19 should be developed with a three dosage regime” requires elaboration so that it is clear why better evidence on models of care is essential in this context. The study findings (discussion section) also need to be considered in the context of COVID-19 if inclusion of these details is going to be meaningful.

There are a number of small revisions that should be considered.

1. Methodology: It would help (for understanding study validity) to briefly summarise in the text the details on non-response that are included in Figure 1. How many people were seen at the relevant services (OAT/CCC) during the study recruitment period, what proportion were eligible to participate and of these how many actually participated? Did those who participated differ in any significant way from those who refused or were unavailable? (If you can distinguish between non-availability and ineligibility). There are a reasonable number who declined consent, so it would help to know if they had different characteristics. What proportion overall were lost to follow up? 

Design and limitations: 

2. Hospital clinics often differ from specialised (e.g. OAT) and community based clinics in important ways other than the range services provided on-site, e.g. service culture/philosophy and social environment (e.g. stigma). The physical environment may also differ. It was noted in the methods section that people randomised to the hospital outpatient condition had to travel independently to the hospital site. People who inject drugs have sometimes reported that hospital clinics are more difficult to access than community based clinics, partly because these settings seem more formal and daunting. It should be considered in the study limitations that some of the group differences could be explained by the study settings. 

3. On a more nuanced level, there is also the question of the intervention elements, i.e. integration and multidisciplinary care. To what extent are the treatment effects attributable to the easy on site access for OAT clients treated at an integrated service, rather than the additional support and engagement offered by a multidisciplinary team? More detail about the composition of the teams (in the context of this study) and the services they provided would be helpful. It may not necessarily be possible to fully address the role of the different intervention elements, but the question should at least be considered.

4. In the discussion it would be worthwhile briefly reflect on the limitations of 12 week SVR as a measure of treatment completion and cure. Rates of SVR testing in PWID are often around 80-85% but are unlikely to represent true outcomes. 

5. The written expression is generally OK but the paper could use a thorough proof read. There are some errors in grammar and expression, e.g. Abstract: "The aim of the study was to evaluates the efficacy of integrated treatment of HCV infection among PWID with standard treatment" and "Among 298 included participants people randomized to standard treatment" and "Integrated treatment for HCV in was superior to….." and in the Discussion (p. 9; spelling) "Among those who initiated treatment in our trial, 84% achieved SVR which is slightly fewer compared to a trial on direct acting antiviral medications were 92% achieved SVR" ('were' should be 'where').

6. Page 5: "…eligible for treatment according to national HCV treatment guidelines". Is it possible to provide a reference to these guidelines (or append)?

7. Randomization and masking: Please briefly specify the masking measures within this paper; this an important detail and the reader shouldn't need to refer to another paper for this.

8. You should consider noting the treatment side effects in the discussion because these have implications for client education, engagement, and treatment management. 

Andrew Smirnov, The University of Queensland.

[LINK]

---

## [Decision Letter · Decision Letter 2]

11 Mar 2021

Dear Dr. Fadnes,

Thank you very much for re-submitting your manuscript "Integrated treatment of hepatitis C virus infection among people who inject drugs: A multi-center randomized controlled trial (INTRO-HCV)" (PMEDICINE-D-20-04289R2) for review by PLOS Medicine.

I have discussed the paper with my colleagues and it was also seen again by reviewers 1-2. I am pleased to say that provided the remaining editorial and production issues are dealt with we are planning to accept the paper for publication in the journal.

[LINK]

We look forward to receiving the revised manuscript by Mar 18 2021 11:59PM.   

Sincerely,

Dr Raffaella Bosurgi, 

Executive Editor 

PLOS Medicine

plosmedicine.org

Requests from Editors:

Comments from Reviewers:

Reviewer #1: The revisions address the concerns raised and the paper is suitable for publication.

Minor points:

The addition of fibroscan data in table 1 is interesting in that 1/3 of patients had fibrosis or cirrhosis, a high figure but consistent with previous reports. However, please check the asterisks to link up the definitions for 'fibrosis' and 'cirrhosis' as i think these have been mixed up.

Reviewer #2: Alex McConnachie, Statistical Review

I thank the authors for their consideration of my original comments. I am happy with almost all of their responses. The only thing that I feel is still lacking are interaction tests for the subgroup analyses presented in Figure 3 and 4. All that would be required is a column of interaction p-values down the right hand side of each figure. Less importantly, each group of estimates and CIs could be separated slightly from the others - this would make it easier to follow

[LINK]

---

## [Decision Letter · Decision Letter 3]

10 May 2021

Dear Dr Fadnes, 

On behalf of my colleagues and the Academic Editor, Dr. Mirjam E. E. Kretzschmar, I am pleased to inform you that we have agreed to publish your manuscript "Integrated treatment of hepatitis C virus infection among people who inject drugs: A multi-center randomized controlled trial (INTRO-HCV)" (PMEDICINE-D-20-04289R3) in PLOS Medicine.

PRESS

Sincerely, 

Beryne Odeny 

Associate Editor 

PLOS Medicine